# Expert-augmented automated machine learning optimizes hemodynamic predictors of spinal cord injury outcome

Austin Chou[1,2,3], Abel Torres-Espin[1,2,3], Nikos Kyritsis[1,2,3], J. Russell Huie[1,2,3], Sarah Khatry[4], Jeremy Funk[4], Jennifer Hay[4], Andrew Lofgreen[4], Rajiv Shah[4], Chandler McCann[4], Lisa U. Pascual[5], Edilberto Amorim[3,6], Philip R. Weinstein[2,6,7], Geoffrey T. Manley[1,2,3], Sanjay S. Dhall[1,2,3], Jonathan Z. Pan[1,8], Jacqueline C. Bresnahan[1,2,3], Michael S. Beattie[1,2,3], William D. Whetstone[9], Adam R. Ferguson[1,2,3]*, the TRACK-SCI Investigators[¶]

1 Weill Institute for Neurosciences, Brain and Spinal Injury Center, University of California, San Francisco (UCSF), San Francisco, California, United States of America, 2 Department of Neurological Surgery, University of California, San Francisco (UCSF), San Francisco, California, United States of America, 3 Zuckerberg San Francisco General Hospital and Trauma Center, San Francisco, California, United States of America, 4 DataRobot, Inc., Boston, Massachusetts, United States of America, 5 Orthopedic Trauma Institute, Department of Orthopedic Surgery, University of California, San Francisco (UCSF), San Francisco, California, United States of America, 6 Department of Neurology, University of California, San Francisco (UCSF), San Francisco, California, United States of America, 7 Weill Institute for Neurosciences, Institute for Neurodegenerative Diseases, Spine Center, University of California, San Francisco (UCSF), San Francisco, California, United States of America, 8 Department of Anesthesia and Perioperative Care, University of California, San Francisco (UCSF), San Francisco, California, United States of America, 9 Department of Emergency Medicine, University of California, San Francisco (UCSF), San Francisco, California, United States of America

¶ TRACK-SCI Consortia authorship. Full author list in Acknowledgments.
* Adam.Ferguson@ucsf.edu

**Data Availability Statement:** The training and validation datasets analyzed in the current study will be published and made publicly available in the Open Data Commons for Spinal Cord Injury (ODC-

## Abstract

Artificial intelligence and machine learning (AI/ML) is becoming increasingly more accessible to biomedical researchers with significant potential to transform biomedicine through optimization of highly-accurate predictive models and enabling better understanding of disease biology. Automated machine learning (AutoML) in particular is positioned to democratize artificial intelligence (AI) by reducing the amount of human input and ML expertise needed. However, successful translation of AI/ML in biomedicine requires moving beyond optimizing only for prediction accuracy and towards establishing reproducible clinical and biological inferences. This is especially challenging for clinical studies on rare disorders where the smaller patient cohorts and corresponding sample size is an obstacle for reproducible modeling results. Here, we present a model-agnostic framework to reinforce AutoML using strategies and tools of explainable and reproducible AI, including novel metrics to assess model reproducibility. The framework enables clinicians to interpret AutoML-generated models for clinical and biological verifiability and consequently integrate domain expertise during model development. We applied the framework towards spinal cord injury prognostication to optimize the intraoperative hemodynamic range during injury-related surgery and additionally identified a strong detrimental relationship between intraoperative hypertension and patient outcome. Furthermore, our analysis captured how evolving clinical

SCI.org) repository concurrent with the publication of the manuscript. The associated dataset DOI(s) directing to the published datasets will be provided and this section updated prior to manuscript publication.

**Funding:** This work was supported by research grants: Department of Defense/Congressionally Directed Medical Research Programs/Spinal Cord Injury Research Program SC150198 (MSB) and SC190233 (MSB); Craig H. Neilsen Foundation SCI - Center of Excellence Award (MSB); National Institute of Health/National Institute of Neurological Disorders and Stroke R01NS088475 (ARF) and UH3NS106899 (ARF); US Department of Veterans Affairs 1I01RX002245 (ARF); US Department of Veterans Affairs I01RX002787 (ARF); Craig H. Neilsen Foundation (ARF); Wings for Life Foundation (ARF); National Institute of Health/ National Institute of Neurological Disorders and Stroke National Research Service Award F32NS117728 (AC). Authors AC, SK, JF, JH, AL, RS, and CM are current or former employees of DataRobot and own shares of the company. Authors SK, JF, JH, AL, RS and CM received support in the form of salaries from DataRobot. The funders had no role in study design, data collection and analysis, decision to publish, or preparation of the manuscript. DoD/CDMR/SCIRP: https://cdmrp.army.mil/scirp/default Craig H. Neilsen: https://chnfoundation.org/ NIH/NINDS: https://www.ninds.nih.gov/ US VA: https://www.research.va.gov/ Wings for Life: https://www.wingsforlife.com/us.

**Competing interests:** I have read the journal's policy and the authors of this manuscript have the following competing interests: AC, SK, JF, JH, AL, RS, and CM are current or former employees of DataRobot and own shares of the company. Access to the DataRobot Automated Machine Learning platform was awarded through application and selection by the DataRobot AI for Good program. DataRobot affiliated authors provided editorial contributions during the preparation of the manuscript. All other authors have declared that they have no competing interests.

practices such as faster time-to-surgery and blood pressure management affect clinical model development. Altogether, we illustrate how expert-augmented AutoML improves inferential reproducibility for biomedical discovery and can ultimately build trust in AI processes towards effective clinical integration.

## Introduction

Automated machine learning (AutoML) is a rapidly-developing ML subfield focused on automating model optimization processes including algorithm selection, feature engineering, and hyperparameter tuning [1, 2]. AutoML applications produce high-performance models across diverse sophisticated algorithms and preprocessing methodologies while reducing the overall need for human input and modeling expertise [3, 4]. Correspondingly, AutoML is lowering the technical and knowledge barrier impeding ML democratization for various domains including biomedicine [5–9]. With the growing popularity of artificial intelligence and machine learning (AI/ML) in clinical research [10–13] and the increasing breadth, depth, and accessibility of clinical health data [14], AutoML stands to exponentially accelerate clinical ML applications by empowering scientists and clinicians to train and leverage powerful models [5]. However, clinical utility requires ML models to be interpretable for biological mechanisms, verifiable by clinicians, and methodologically and inferentially reproducible [15, 16]. Achieving reproducibility is further complicated by the fact that clinical datasets often have small sample sizes relative to the number of variables collected which can result in unstable model behavior [17, 18]. This is especially true for rare diseases and disorders with smaller patient populations, and translation of AutoML from computer-to-clinic thus necessitates additional approaches beyond maximizing prediction accuracy with "black box" algorithms. Here, we developed a modeling framework that incorporates explainable and reproducible AI strategies to predict spinal cord injury (SCI) patient outcome. Furthermore, we demonstrate how we can improve the inferential reproducibility of ML, augment the process with clinical knowledge, and effectively leverage AutoML for model optimization.

While SCIs have a comparatively small patient population—about 17,900 new cases a year and 296,000 patients with chronic disabilities in the US—SCIs are highly debilitating and result in chronic motor, sensory, and autonomic impairment including paralysis [19]. The severity is reflected by the total societal cost which is estimated to exceed $267 billion [20]. Alongside the relatively limited sample size of available clinical SCI datasets, the variety of SCI characteristics presents significant challenges for reproducible identification of patient outcome predictors despite the volume of data collected throughout patient hospitalization and treatment [20]. Various prognostic models for SCI outcome have been developed with algorithms ranging from logistic regression to extreme gradient boosted (XGB) trees and convolutional neural networks [12, 21, 22]. While such studies bear potential for informing clinical care, algorithm selection in many SCI ML studies have primarily depended on the researchers' familiarity with specific ML algorithms, and prediction accuracy remains the primary metric for comparing models [23, 24]. Moreover, deciphering the relationships between outcome and predictors is still difficult with complex algorithms, ultimately dampening clinician enthusiasm about applying ML tools and results given the inability to interpret and verify such models [25].

In our AutoML application for SCI patient prognosis, we demonstrate:

- A framework for reproducible and explainable modeling that implements (1) a repeated cross-validation strategy, (2) *performance precision* and *feature instability* metrics and

AutoML framework for inferential reproducibility in biomedicine

**Fig 1. A framework for applying Automated Machine Learning (AutoML) for reproducible inferences in biomedical research.** After data is curated, we perform a cyclical model development process utilizing AutoML to optimize an array of models. Reproducible and explainable AI tools and strategies can be applied to ultimately draw clinical and biological inferences from the models and allow for integration of domain expertise. Critically for clinical modeling, we also include a feature reduction component to achieve a more parsimonious model. The final models are then validated with external validation data along with population similarity analysis for further clinical contextualization. By applying this framework, models produced by AutoML can be stabilized and interpreted for inferential reproducibility and clinical verifiability.

analyses, (3) model interpretation with permutation feature importance (pFI) and partial dependence plots (PDPs), (4) stabilized backward feature reduction, and (5) model validation with population similarity analysis (Fig 1). In particular, repeated cross-validation allows for model aggregation to account for modeling variability and improve the inferential reproducibility of the results.

- The importance of integrating domain expertise. We highlight how stabilized pFI and PDPs, useful model-agnostic explainable AI tools, enable biomedical researchers to draw robust inferences regarding the relationship between clinical variables and outcome. Furthermore, we illustrate how augmenting feature selection with domain expertise can improve model performance beyond deploying ML naively.

- Additional analyses to interpret model validity. Many clinical ML studies with smaller sample sizes depend on newly collected data for external validation. Investigating the population similarity between training and validation cohorts provides meaningful information about model generalizability beyond validation performance and can capture evolving clinical practices that inevitably affects clinical ML implementation.

By applying this framework to SCI, we identified actionable intraoperative mean arterial pressure (MAP) thresholds for hypertension and hypotension associated with worse patient outcome. Additionally, our analysis revealed underlying evolutions of clinical practices, such

as reducing time-to-surgery and hypotension management, that invariably affects model validation efforts in SCI research. Altogether, we present methods to bolster the interpretability, reproducibility, and trustworthiness of clinical ML especially as AI/ML and AutoML becomes increasingly accessible to biomedical researchers.

## Results

### AutoML model generation

We applied an AutoML platform to investigate clinical predictors of SCI patient outcome from intraoperative and acute hospitalization records collected between 2005–2011 and curated by the Transforming Research and Clinical Knowledge for SCI (TRACK-SCI) program, one of the largest SCI patient registries in the US [20]. We selected 46 variables (i.e. features) as predictors from de-identified data of 74 patients (S1 Table). Of these, 16 features were summary statistics (i.e. mean, standard deviation, skew, and kurtosis) derived from timeseries data capturing heart rate, systolic blood pressure, diastolic blood pressure, and mean arterial pressure (MAP) during SCI surgery. As intraoperative hypertension and hypertension have been shown to be detrimental to SCI outcome [26, 27], we also calculated the time each patient spent outside of previously-established upper (104 mmHg) or lower (76 mmHg) MAP thresholds during surgery (*time_MAP_Avg_above_104* and *time_MAP_Avg_below_76*, respectively) [28]. We defined the prediction target as whether the patient's ASIA Impairment Scale (AIS) score, a common SCI outcome assessment [29], improved between time of hospital admission and time of hospital discharge.

To account for potential instability in model optimization, we applied a repeated 10-fold cross-validation strategy with 25 repetitions where each had a unique partitioning arrangement (i.e. 25 projects) [30, 31]. We then aggregated the results for analysis. AutoML generated 80–90 blueprints: unique combinations of data preprocessing methods and ML algorithms. From these, the platform fully optimized 30–40 models, 15 of which had better mean performance (lower LogLoss and higher AUC) than the benchmark majority class classifier model (Fig 2A and 2B).

For the purposes of illustration, we selected two high performance blueprints for further interpretation and validation. The first was a L2 regularized logistic regression model with a spline transformation of numeric variables during data preprocessing (BP$_{log}$; Fig 2C). BP$_{log}$ had the best overall performance by LogLoss ($0.67 \pm 0.01$) and was a top performer by AUC ($0.68 \pm 0.02$). To apply our framework to a highly complex model use case and given the popularity of XGB in biomedical ML research, we also examined the "eXtreme gradient boosted trees classifier with unsupervised learning features" blueprint with the best LogLoss performance in its class (BP$_{XGB}$; LogLoss = $0.68 \pm 0.01$; AUC = $0.67 \pm 0.02$). Importantly, XGB trees have gained popularity in biomedical ML research. BP$_{XGB}$ specifically includes a TensorFlow Variational Autoencoder preprocessing step [32] (Fig 2D) as the "unsupervised learning feature", exemplifying the availability of sophisticated methodologies through AutoML platforms.

### Feature importance

We utilized a permutation-based approach [33] to quantify feature importance (pFI) for BP$_{log}$ and BP$_{XGB}$. Notably, pFI values from individual models varied significantly; we accordingly aggregated pFI across the 25 projects for more robust comparisons. While the order of features by importance for BP$_{log}$ and BP$_{XGB}$ were different, we observed that many of the high importance features for both models were the timeseries summary statistics (S1 Fig). Interestingly, *time_MAP_Avg_above_104* and *time_MAP_Avg_below_76* were the most important features for BP$_{log}$ (S1A Fig) but were 11th and 18th in rank respectively for BP$_{XGB}$ (S1B Fig).

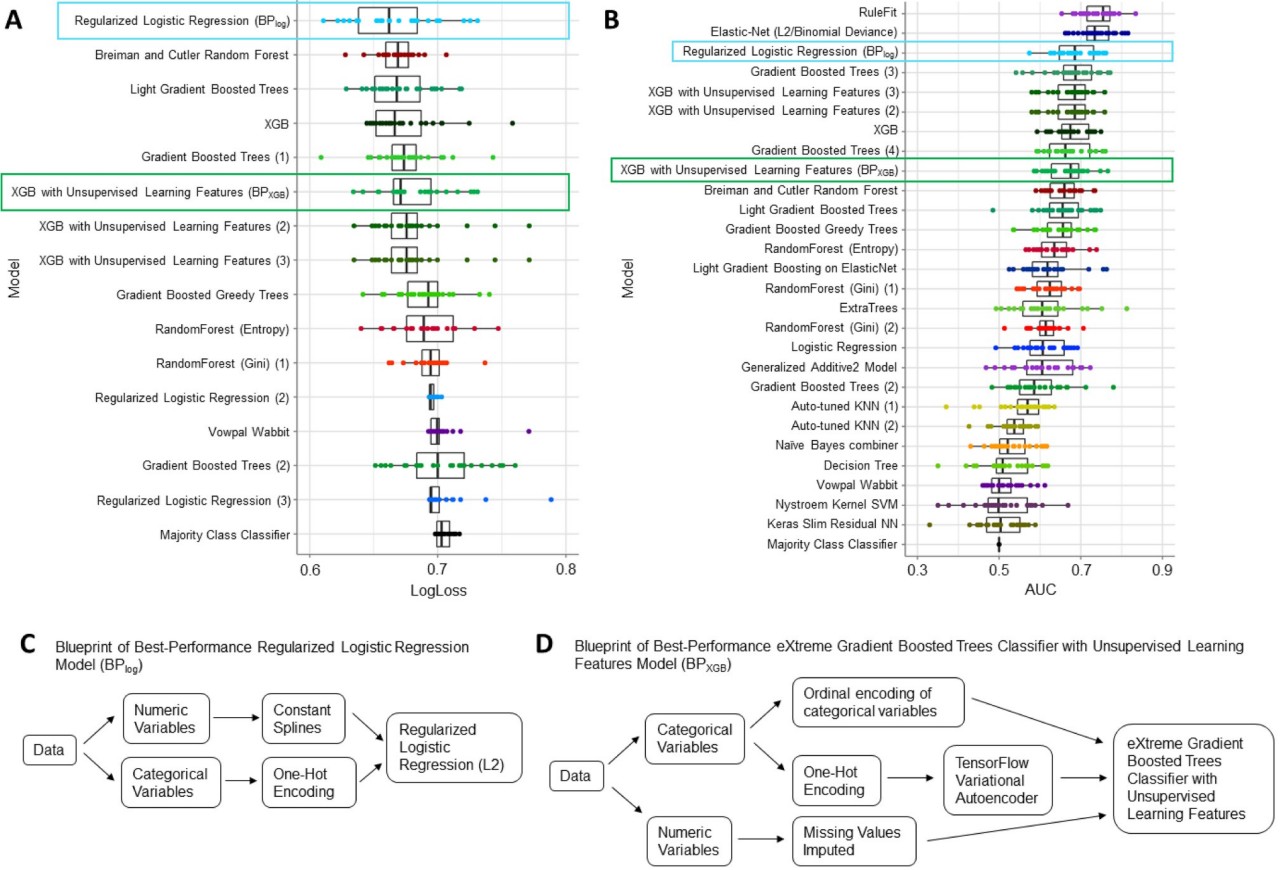

**Fig 2. AutoML generated 15 models that performed better than the Majority Class Classifier model.** Each model consisted of automatically implemented preprocessing steps and algorithms. Models were assigned names according to the algorithm and encoded by a unique color. Blueprints of the same algorithm class are numbered for identification across both (A) LogLoss and (B) Area Under Curve (AUC) plots. Two models were selected for additional analysis: $BP_{log}$ (blue box) and $BP_{XGB}$ (green box). Aggregating across 25 projects (unique partitioning arrangements of the dataset), $BP_{log}$ had an average performance of 0.67 ± 0.01 LogLoss and 0.68 ± 0.02 AUC; $BP_{XGB}$ had an average performance of 0.68 ± 0.01 LogLoss and 0.67 ± 0.02 AUC. (C) $BP_{log}$ consisted of a regularized logistic regression (L2) algorithm with a notable quintile spline transformation preprocessing step for numeric variables. (D) $BP_{XGB}$ implemented an eXtreme Gradient Boosted (XGB) trees classifier with unsupervised learning features, which refers to the TensorFlow Variational Autoencoder preprocessing step for categorical variables.

## Performance precision

We observed that different partitioning arrangements resulted in varying model performances even with the same blueprint. We sought to determine whether aggregating the results of 25 projects significantly improved the precision of model performances (i.e. *performance precision*) towards reproducible comparisons between blueprints. In addition to the confidence intervals (CI) for each blueprint, we additionally defined and calculated the *standardized performance CI width*: a measure of the performance precision relative to the model's mean performance (see Methods for formula). For example, the standardized performance CI width allows us to identify the number of projects needed to obtain a CI width that is within 5% of the blueprint performance. From the 25 projects in our primary repeated cross-validation workflow, we observed standardized CI widths of 1.92% by LogLoss and 2.88% by AUC for $BP_{log}$ and 1.55% by LogLoss and 2.73% by AUC for $BP_{XGB}$.

To investigate how performance precision changes with the number of projects aggregated, we ran 150 projects with $BP_{log}$ and $BP_{XGB}$ and performed a sampling analysis (see Methods).

For $BP_{log}$, we observed that aggregating 25 projects more than halved the expected standardized performance CI width when compared to only 2 projects: by LogLoss, 5.22 ± 0.24% vs 1.85 ± 0.01% for 2 projects vs 25 projects respectively (S2A Fig) and by AUC, 8.03 ± 0.38% vs 2.79 ± 0.02% respectively (S2B Fig). Analysis of $BP_{XGB}$ performance precision generated similar results: by LogLoss, 5.42 ± 0.32% vs 2.06 ± 0.04% for 2 projects vs 25 projects respectively (S2C Fig) and by AUC, 8.36 ± 0.38% vs 2.84 ± 0.02% respectively (S2D Fig). The performance precision analysis further highlights the variability in model performances by different partitioning arrangements even if the blueprint and training dataset are unchanged.

## Feature instability

We similarly observed that different partitioning arrangements resulted in pFI variability (i.e. *feature instability*). Given two different pFI lists—for example, from two different modeling projects or multi-project aggregates with corresponding averaged pFI—we can quantify the differences between them by calculating the *feature rank instability (FRI)* (see Methods for formula). The FRI value sums the difference in the pFI-based ranking for each feature shared between any two pFI lists. Higher FRI indicates more dissimilarity in ranking and thus more feature instability.

Similar to the performance precision analysis, we performed a sampling analysis with 150 projects to determine the relationship between number of projects aggregated and feature instability for $BP_{log}$ and $BP_{XGB}$ (see Methods). We observed extremely high FRI when the number of aggregated projects is small, suggesting that pFI can differ significantly from one project to another. When we increased the number of aggregated projects towards 150, FRI decreased towards 0, indicating that pFI ranking can be stabilized with sufficient project aggregation. At 25 projects, $BP_{log}$ and $BP_{XGB}$ had average FRI values of 13.03 ± 0.34 and 11.65 ± 0.33 respectively (Fig 3A and 3B). This approximately amounted to a 93% decrease in instability for both $BP_{log}$ and $BP_{XGB}$ as compared to when only aggregating across 2 projects.

## Automated feature reduction

A component of AutoML is automating feature reduction (i.e. variable selection) to obtain a more parsimonious feature list [34, 35]. This is particularly important for clinical models since clinical features often outnumber the observations in biomedical datasets, increasing the danger of model overfitting [36]. We employed an iterative backward wrapper approach utilizing pFI to determine and remove the lowest-importance features at each step (see Methods).

Importantly, we needed to ascertain the stability of features to be eliminated. Because our approach initially removes five features at a time, we accordingly applied our feature instability analysis to just the five lowest-importance features. With 25 projects, we found that $BP_{log}$ had a FRI value of 0.96 ± 0.08 (Fig 3C) and $BP_{XGB}$ had a FRI value of 0.56 ± 0.06 (Fig 3D) for the bottom five features. In contrast, the bottom five features cumulatively shifted at least eight ranks on average if we aggregated only two projects. By aggregating across 25 projects, we can be confident that the least important features are reliably the lowest ranked.

The best-performing parsimonious $BP_{log}$ had an average LogLoss of 0.55 ± 0.02 and only nine retained features (Fig 4A). Of these, the highest pFI features included *time_MAP_Avg_above_104*, *time_MAP_Avg_below_76*, and the MRI BASIC score (*MRI_1_BASIC_Score*), a neuroimaging score for injury severity collected upon hospital admission (Fig 4B) [37]. The corresponding mean AUC (0.83 ± 0.02) was also close to the maximum AUC of the feature-reduced models (S3A Fig).

Interestingly, initial feature reduction with $BP_{XGB}$ removed the *time_MAP_Avg_below_76* feature. Given that clinical experts and SCI literature emphasize the correlation between

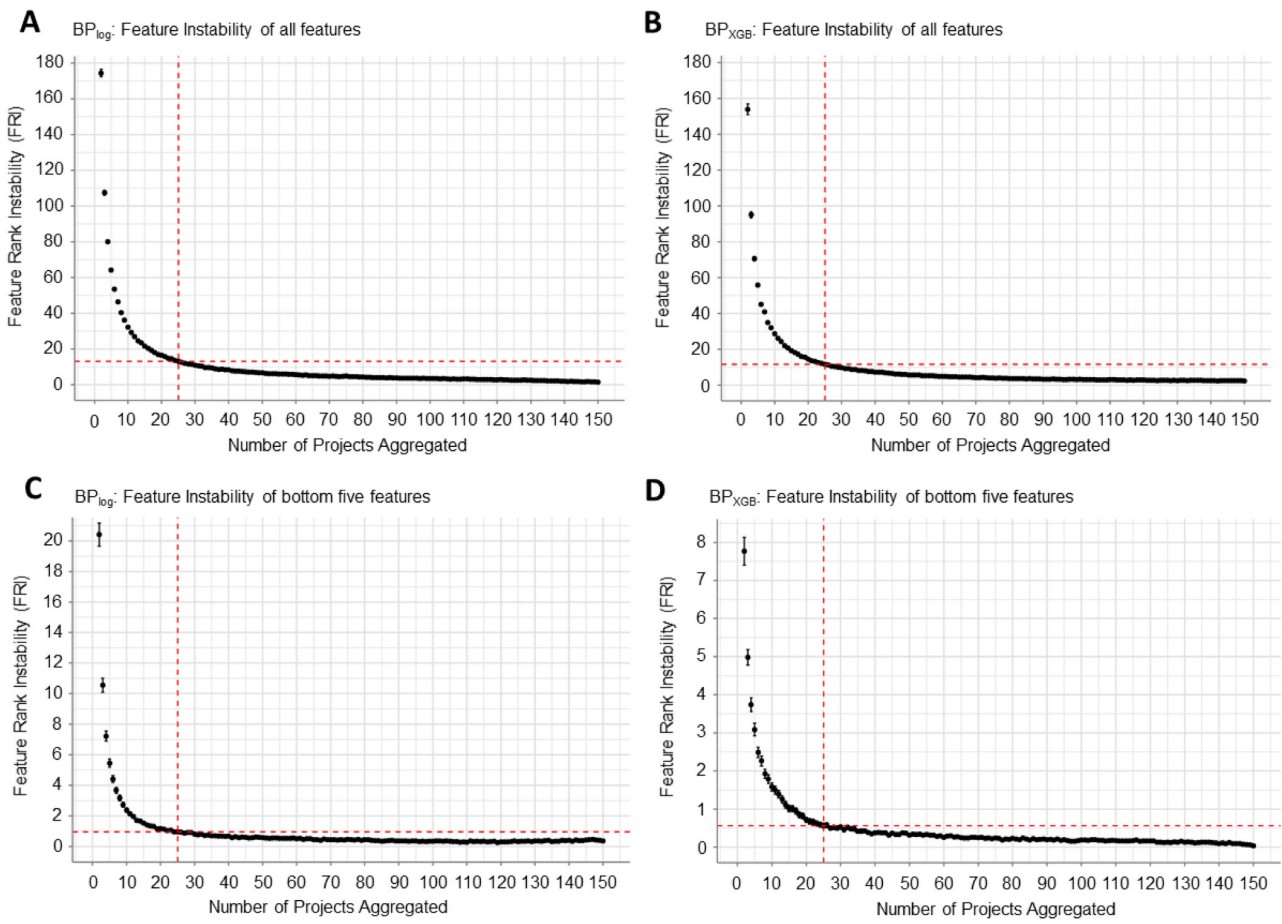

**Fig 3. Feature rank instability (FRI) analysis as a function of number of projects aggregated.** As the number of projects increased, FRI decreased (i.e. pFI ranking became more stable). (A, B) Expected FRI calculated for all 46 features. $BP_{log}$ had an average FRI of 174.40 ± 2.14 with 2-project aggregation and 13.03 ± 0.34 with 25-project aggregation (A). Similarly, $BP_{XGB}$ started with an average FRI of 153.83 ± 3.06 that decreased to 11.65 ± 0.33 at 25 projects (B). (C, D) Focusing only on the bottom five features by pFI to calculate FRI, $BP_{log}$ had an average FRI of 20.41 ± 0.75 with 2-project aggregation and decreased to 0.96 ± 0.08 with 25-project aggregation (C). Similarly, $BP_{XGB}$ started with an average FRI of 7.77 ± 0.37 and decreased to 0.56 ± 0.06 for the bottom five features with 25-project aggregation (D).

hypotension and worse patient outcome [28], we tested if preserving *time_MAP_Avg_below_76* during feature reduction would produce better parsimonious model performance. The resulting parsimonious $BP_{XGB}$ model included 11 features, had an average LogLoss of 0.48 ± 0.02 (Fig 4C), and was close to the maximum AUC observed (0.87 ± 0.01) (S3B Fig). Notably, this final performance was better than the best parsimonious model when *time_MAP_Avg_below_76* had been eliminated (LogLoss = 0.52 ± 0.02; AUC = 0.87 ± 0.01). Furthermore, *time_MAP_Avg_below_76* did not end up as the lowest-ranked feature in the final parsimonious feature list despite requiring user guidance to prevent elimination (Fig 4D). Since *time_MAP_Avg_below_76* exhibited collinearity with the other surgical timeseries-derived features, the handling of collinear features by the XGB algorithm is likely why *time_MAP_Avg_below_76* was dropped without user intervention. Indeed, we observed higher FRI at each feature reduction step for $BP_{XGB}$ as compared to $BP_{log}$ corresponding with larger pFI changes for $BP_{XGB}$ as collinear features are eliminated (S4 Fig). Additionally of interest, the most important feature of the parsimonious $BP_{XGB}$ feature list was the AIS score at admission (*AIS_ad*) which provides similar context for initial injury severity as the MRI BASIC score [37].

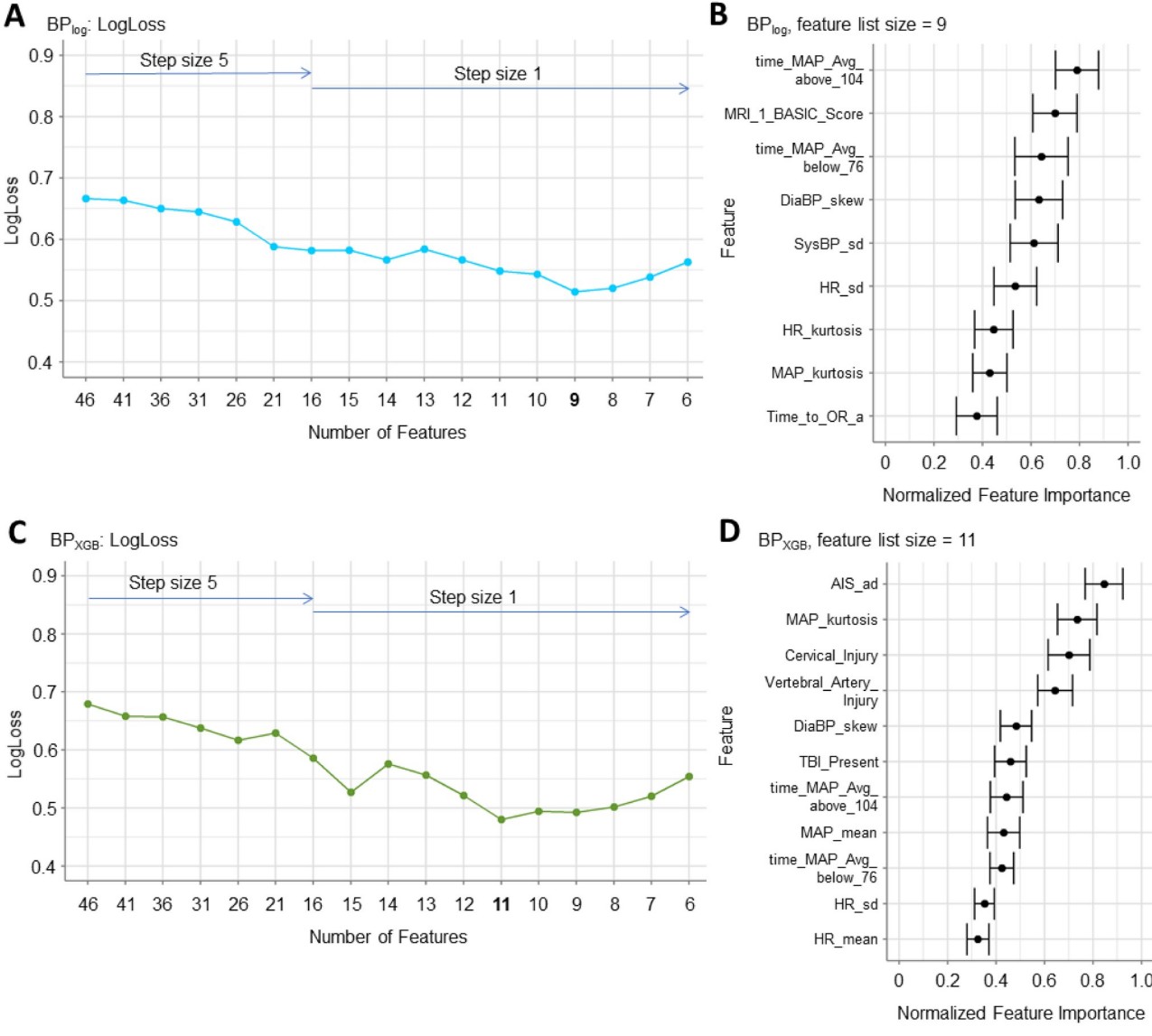

**Fig 4. Applying an iterative backward feature reduction process to identify parsimonious feature lists that maximize model performance.** The process was performed first by removing the lowest five features by feature importance (step size = 5) and then repeated with step size = 1 within the feature list size range that contained the best performance. (A) For BP$_{log}$, the step size was reduced starting at 16 features with the best performance observed with the 9-feature parsimonious feature list (LogLoss = 0.55 ± 0.02). (B) The corresponding pFI of the 9-feature parsimonious BP$_{log}$ model showed that the MRI BASIC score and the time patients spent outside of the MAP thresholds were the most important features. The remaining features included other intraoperative timeseries-derived features and the time between hospitalization and surgery (*Time_to_OR_a*). (C) The feature reduction for BP$_{XGB}$ was expanded to always preserve the two MAP threshold features. The step size was reduced to one starting at 16 features with the best performance observed with the 11-feature parsimonious feature list (LogLoss = 0.48 ± 0.02). (D) The corresponding pFI for the parsimonious BP$_{XGB}$ model showed that the AIS score at admission (*AIS_ad*) was the most important feature. Non-timeseries-derived features included *Cervical_Injury*, *Vertebral_Artery_Injury*, and *TBI_Present*. The *time_MAP_Avg_above_104* and *time_MAP_Avg_below_76* features were ranked 7th and 9th respectively.

## Feature interpretation

For additional interpretability, we utilized partial dependence plots (PDPs) to quantify the relationship of each individual features' values to the model's prediction [38–40]. PDPs are a model-agnostic approach and thus can be applied regardless of the preprocessing steps or

algorithm implemented. We aggregated the partial dependence for each feature for $BP_{log}$ and $BP_{XGB}$ across the 25 projects (S5 and S6 Figs).

The PDPs of the initial injury severity features–*MRI_1_BASIC_Score* for $BP_{log}$ and *AIS_ad* for $BP_{XGB}$–captured nuances between the two (Fig 5A and 5B). We observed that a BASIC score of 4, which corresponds to severe injuries with notable hemorrhage, reduced the probability of patient improvement. Similarly, patients classified as AIS A (i.e. complete, severe SCIs) had the lowest probability of outcome improvement. Both PDPs thus conveyed that the most severe SCI cases are unlikely to see improvement by the time of hospital discharge. However, the two scores had different effects for mild injuries: for $BP_{log}$, low BASIC scores (0–3) all increased the probability of improvement whereas AIS D reduced the likelihood of improvement for $BP_{XGB}$. This underscores the difference in sensitivity and granularity between the AIS and BASIC scores. AIS D broadly encapsulates mild SCIs and is effectively a ceiling on the scale since improvement requires full recovery, which is uncommon. BASIC scores of 0–2 cover a range of functionally mild-to-moderate SCIs; indeed, BASIC and AIS scores do not correlate 1:1 [37]. Overall, the results suggested that patients with moderate SCIs (AIS B and C; BASIC 2) have the highest likelihood of outcome improvement.

PDPs of *time_MAP_Avg_above_104* and *time_MAP_Avg_below_76* revealed that the models predicted worse outcome if a patient exceeded 104 mmHg by more than 70 minutes (Fig 5C and 5D) or dropped below 76 mmHg for more than 150 minutes (Fig 5E and 5F). $BP_{log}$ and $BP_{XGB}$ produced similar *time_MAP_Avg_above_104* and *time_MAP_Avg_below_76* PDPs, though we observed that $BP_{XGB}$ predicted relatively better outcome for patients at the extreme upper range of time (>115 min for *time_MAP_Avg_above_104* and >200 min for *time_MAP_Avg_below_76*). This is likely due to the dataset having fewer patients at the extreme ranges rather than a true clinical effect. Critically, $BP_{log}$ implemented a spline transformation (Fig 2C) and quintiled the continuous variables; all patients exceeding 70 or 150 minutes outside the upper or lower thresholds respectively were categorized to the same quintile. Accordingly, $BP_{log}$ would not produce the PDP rebound observed with $BP_{XGB}$.

## MAP threshold validation

We previously found that time outside the MAP range of 76–104 mmHg was associated with lower probability of AIS improvement as determined by LASSO logistic regression models testing different MAP ranges while expanding the lower and upper MAP thresholds simultaneously [28]. To validate the MAP thresholds, we started with the best-performing parsimonious feature lists, removed the MAP threshold features, and then swept through various lower threshold (70–85 mmHg) or upper threshold (95–115 mmHg) features separately to identify how the threshold affected prediction.

With $BP_{log}$, we observed the best performances with lower thresholds of 74–76 or 79 mmHg and with upper thresholds at 103–105 mmHg by LogLoss (Fig 6A). With $BP_{XGB}$, we observed the best performances at lower thresholds of 74–76 mmHg and at upper thresholds at 103–104 mmHg (Fig 6B). The results were similarly reflected with AUC (S7 Fig), corroborating the thresholds of 76 and 104 mmHg for predicting patient outcome. Our analysis furthermore revealed that the time spent above the upper threshold improved the predictive performance of the models significantly more, highlighting intraoperative hypertension as an important correlate and a potential factor for worse SCI recovery.

## Model validation

We visualized how each step of the workflow improves AUC and corresponding receiver operating characteristic (ROC) curves. Starting with $BP_{log}$ and all our predictors except MAP

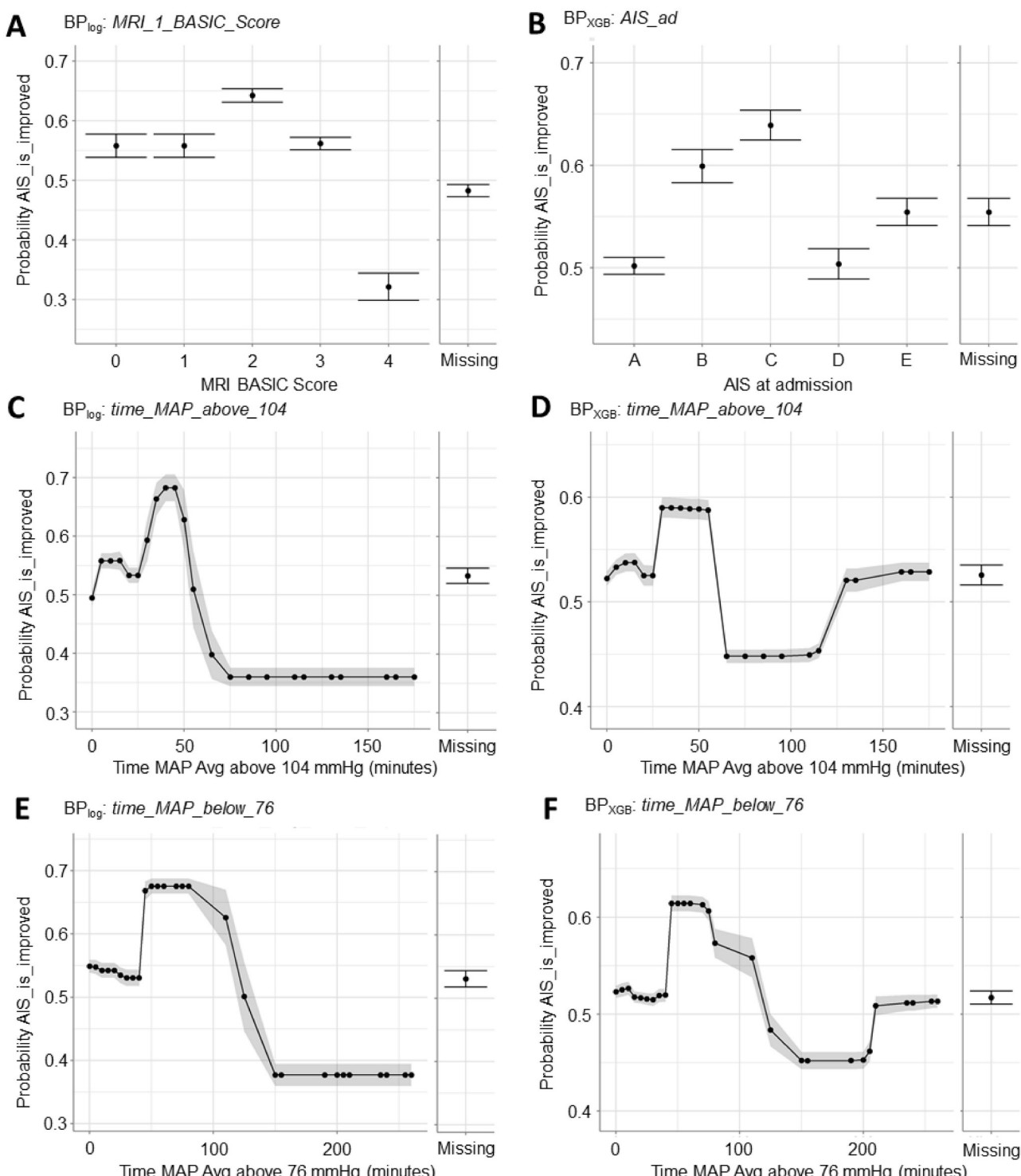

**Fig 5. Partial dependence plots (PDPs) for features of interest help interpret how features affect model prediction of BP$_{log}$ and BP$_{XGB}$.** (A) For BP$_{log}$, an MRI BASIC score of 4 resulted in lower prediction of improved outcome. A MRI BASIC score of 0–3 increased prediction of better outcome with a MRI BASIC score of 2 leading to the highest probability of improvement. (B) For BP$_{XGB}$, an AIS score of A or D at admission resulted in lower probability of patient improvement. AIS scores of B and C both led to higher probability of improvement with AIS score C resulting in the highest probability. (C) For BP$_{log}$ and (D) BP$_{XGB}$, if a patient's MAP exceeded an upper threshold of 104 mmHg for more than 50–75 minutes, the predicted probability of improvement decreased significantly. (E) For BP$_{log}$ and (F) BP$_{XGB}$, if a patient's MAP fell below a lower threshold of 76 mmHg for more than 100–150 minutes, the predicted probability of improvement decreased significantly. Notably, BP$_{XGB}$ PDP for both *time_MAP_Avg_above_104* and *time_MAP_Avg_below_76* exhibited a rebound in predicted improvement probability at extreme upper values that was absent on the BP$_{log}$ PDPs.

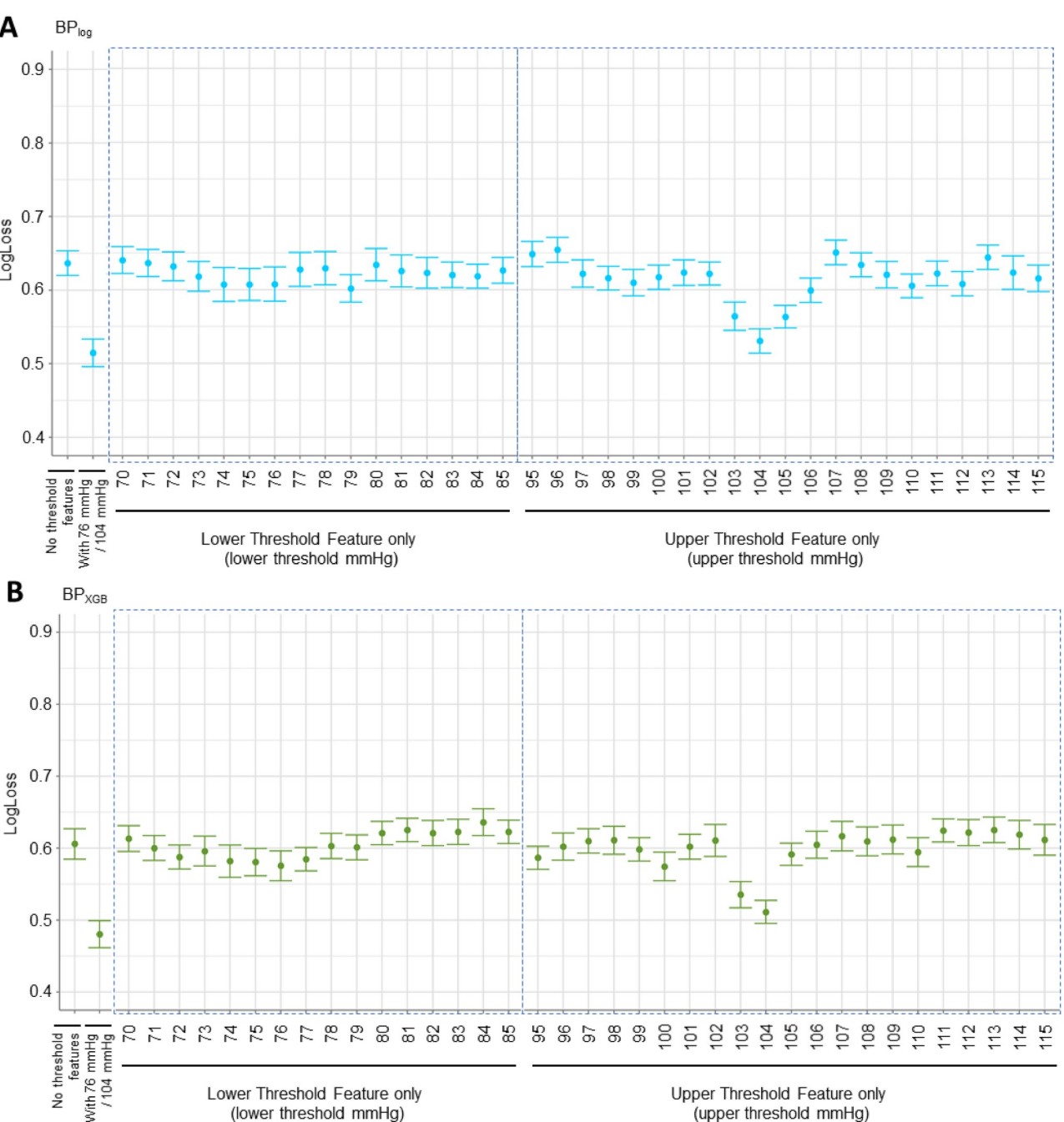

**Fig 6. LogLoss performance plots for investigating different lower and upper MAP thresholds using best-performing parsimonious BP$_{log}$ and BP$_{XGB}$ models.** (A) With BP$_{log}$, we observe that the lower threshold values of 74, 75, 76, and 79 mmHg performed the best of the lower thresholds. The upper threshold values of 103, 104, and 105 mmHg performed the best of the upper thresholds. Notably, the best-performing upper threshold feature (104 mmHg) resulted in a larger improvement to model performance compared to the best-performing lower threshold feature (79 mmHg). (B) With BP$_{XGB}$, the values of 74, 75, and 76 mmHg performed the best of the lower thresholds, and the values of 103 and 104 performed the best of the upper thresholds. Similar to BP$_{log}$, the best-performing upper threshold feature (104 mmHg) resulted in a larger improvement to model performance compared to the best-performing lower threshold feature (76 mmHg).

threshold features, we achieved an average AUC of 0.63 ± 0.019 (S8A Fig). Inclusion of MAP threshold features (with 76 and 104 mmHg thresholds) improved AUC to 0.68 ± 0.02 (S8B Fig), underscoring the importance of intraoperative MAP regulation. The largest improvement to model performance occurred after feature reduction to a more parsimonious feature list (AUC 0.84 ± 0.02) (S8C Fig). Adjusting the MAP thresholds produced miniscule improvements to AUC (AUC 0.85 ± 0.02) (S8D Fig). Lastly, by repeating the process with $BP_{XGB}$, prediction performance was improved to AUC 0.87 ± 0.01 (S8E Fig). Altogether, we can improve model performance by adjusting the feature list and model by leveraging the AutoML workflow (S8F Fig). Importantly, the ROC curves from each of the 25 projects notably differ, emphasizing how varying the dataset partitioning can produce significant model variability.

Critically, the greatest obstacle for translating predictive models into the clinic is the validity of the models on novel data. While many ML scenarios implement a holdout partition from the original dataset for validation, clinical datasets often have relatively small sample sizes where such practices would result in underfitted models, especially for medical fields with smaller patient populations such as SCI [41]. Clinical model validity is thus often assessed with an external validation dataset collected from a new cohort of patients. Here, we obtained additional data for external validation from a prospective (2015 onward) TRACK-SCI cohort of 59 patients. Of these, 14 patients improved in outcome while the remainder 45 did not.

We used the parsimonious $BP_{log}$ and $BP_{XGB}$ models to predict the probability of AIS improvement of the validation cohort. We aggregated these values across the 25 projects and generated plots and confusion matrices using the mean predictions and best F1 thresholds calculated by the AutoML platform. The 9-feature parsimonious $BP_{log}$ model correctly predicted 13 of the 14 patients who improved but only 15 of the 45 patients who did not (Fig 7A; S9A Fig). The 11-feature parsimonious $BP_{XGB}$ model correctly predicted only 9 of the 14 patients with AIS improvement and 14 of the 45 patients without (Fig 7B; S9B Fig). While $BP_{XGB}$ has higher predictive accuracy on the training dataset, the model did not perform as well on novel data as the $BP_{log}$ model.

We hypothesized that the poor validation performance was due to data drift where the validation patient population no longer resembled that of the training dataset. We accordingly performed population similarity analysis, starting with population stability index (PSI) assessment for each of the parsimonious features. PSI broadly reflects the differences in value distribution between the two cohorts [42]. We observed that most of the features exhibited significant (PSI > 0.25) or moderate (PSI > 0.1) drift between training and validation datasets, and only *TBI_Present* and *Vertebral_Artery_Injury* features could be considered to not have drifted (PSI < 0.1) (S2 Table). The PSI results overall suggested that the training and validation populations are dissimilar, thus resulting in poor model performance during validation.

To investigate clinical trends underlying data drift, we clustered all the patients by the raw feature values of the 15 parsimonious features via UMAP and HDB clustering. We observed notable differences in cluster representation: in particular, the validation cohort was only sparsely represented in Clusters 1 and 2 as compared to the training cohort (Fig 7C). We summarized the distribution of values within each cluster to better understand the subpopulation characteristics and found that Cluster 1 was defined by extremely high *Time_to_OR* and Cluster 2 by extremely high *time_MAP_Avg_below_76* (S3 Table). In discussion with clinical experts, we found that this corresponded with shifting clinical practices to reduce time-to-surgery and prevent intraoperative hypotension [43, 44]. Population similarity analysis thus provided critical insight into the differences between training and validation populations while demonstrating the crucial need to validate predictive models and corresponding conclusions before translating findings to the clinic.

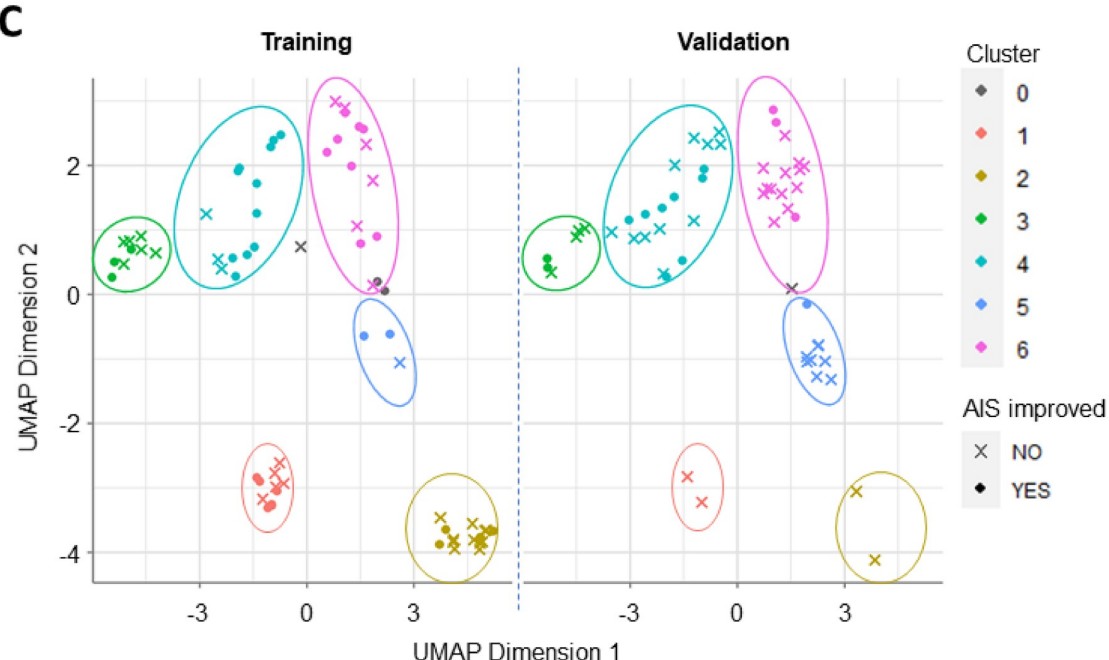

**Fig 7. Model validation confusion matrices and clustering analysis to demonstrate differences in patient population between training and validation datasets.** Validation predictions were scored by comparing the average predicted probability of each validation sample against the average best F1 threshold for the corresponding model. (A) The best parsimonious $BP_{log}$ model correctly predicted 13 of the 14 true positives (i.e. patient improved in outcome) and 15 of the 45 true negatives. (B) The best parsimonious $BP_{XGB}$ model correctly predicted 9 of the 14 true positives and 14 of the 45 true negatives. (C) UMAP and HDB clustering analysis on the combined training and validation data produced six clusters of patients. Notably, Clusters 1 and 2 showed high representation in the training cohort and low representation in the validation cohort. Conversely, Cluster 3 showed low and high representation in the training and validation cohorts respectively. Clusters 3, 5, and 6 have no discernable differences between cohorts.

Overall, we illustrate a framework to augment AutoML for interpretability, reproducibility, and validity. The process presents opportunities to integrate methods for explainable and reproducible AI that are essential for biological inferences and evidence-based clinical practice. Augmenting AutoML to generate verifiable machine intelligence will be critical to building confidence and trust in powerful AI tools towards advancing biomedical research and precision medicine.

## Discussion

As AutoML become increasingly efficient at optimizing an ever-growing repertoire of data preprocessing methods and algorithms, biomedical researchers will have more opportunities to leverage AI/ML to transform their field [4, 6, 9, 45]. Importantly, there remains a parallel, fundamental need for better evaluation and interpretation of AutoML-derived models for healthcare where trust in ML requires reproducibly identifying and validating biomedical inferences before translation to patient care [46]. Here, we demonstrate a framework to

## Box 1. Key highlights of the framework for reproducible, interpretable AutoML application in biomedicine

- Perform modeling with a **repeated cross-validation strategy**. Aggregating across models mitigates spurious findings due to model variance from implicit modeling parameters that can lead to model instability with smaller clinical datasets, such as partitioning arrangement.

- **Stabilize pFI and PDPs** for model interpretation by aggregating repeated cross-validation models to improve inferential reproducibility.

- Characterize **performance precision** (metrics: performance CI; standardized performance CI width) for more robust model comparisons. Performance precision analysis can be further applied to achieve a target precision in modeling processes.

- Characterize **feature instability** (metric: pFI CI; feature rank instability) to capture pFI variability. Feature instability analysis can be further applied to stabilize pFI-dependent processes such as feature reduction.

- **Integrate domain expertise** throughout; combining domain expertise and model-driven feature selection can improve final parsimonious model performance. Any inferences drawn from modeling should be verified with clinical expertise.

- Investigate data drift (i.e. **population similarity analysis**) between the training and external validation datasets during model validation. PSI and clustering analysis can uncover clinical differences between the cohorts and the underlying evolution of clinical practices that further inform model validation results.

improve the reproducibility of ML and AutoML workflows by implementing a repeated cross-validation strategy, assessing metrics of model stability, directly interrogating and augmenting the models with clinical knowledge, and applying population similarity analysis to better contextualize the findings and generalizability of the models (Box 1; Fig 1). By incorporating such methods and strategies for model interpretability and inferential reproducibility, biomedical experts will be empowered to fully leverage their domain expertise into ML processes to construct more trustworthy models and build confidence towards translatable AI/ML applications.

Repeated cross-validation is an easily applied strategy to reinforce the reproducibility of ML research and has been well-established for handling variation in partitioning [47, 48]. Yet despite the focus on confidence intervals, significance, and inference in biomedical research, repeated cross-validation is not commonly implemented by clinical ML studies. This is likely due to the associated computational costs, resulting in clinical ML researchers optimizing and drawing conclusions from a single model instead. We show that different dataset partitions affected model performance and pFI values; the practice of only relying on a single optimization can thus lead to false assumptions that one algorithm is definitively better at prediction or that specific features are definitively more important. Additionally, combining repeated optimizations can help stabilize subsequent processes; 25 projects was the critical aggregation

threshold for our backward feature reduction approach such that the bottom 5 features by pFI would be reproducibly ranked as the least important. Naturally, selecting the number of repetitions ultimately depends on the dataset, the specific model, and the research question or ML process of interest. Nevertheless, a repeated k-fold cross-validation strategy to report confidence intervals, performance precision, and feature instability can help contextualize the reproducibility of results and mitigate spurious conclusions whether applying AutoML or a single model blueprint.

The variability we observed also reflects the concept of underspecification in ML: that even with the same model blueprint and training data, different optimizations can produce divergent solutions [49]. Underspecification highlights how seemingly arbitrary modeling choices —such as implicit modeling parameters and data partitioning—can lead to models with high accuracy on training data that then fail to perform on novel data. While this emphasizes the need to validate models, underspecification also underscores an ongoing demand for additional model evaluation metrics. Repeated cross-validation strategies along with performance precision and feature instability analyses can be applied to characterize and control for underspecification factors of model optimization. This can be further extended to deployment applications: bagging models to account for underspecification can improve model validity, and precision and instability analyses can help estimate the effective number of models for ensembling. Moreover, the predictions of the individual models underlying an ensemble can provide useful context about model precision to users and ultimately enhance trustworthiness and adoption for clinical decision-making.

A major challenge in clinical modeling is obtaining external validation data, especially for rarer diseases and disorders [20, 41]. Inconsistent data collection methods and standards further exacerbate the difficulties. Many clinical ML studies consequently produce accurate models that remain unvalidated or are later shown to underperform on novel data [50–52]. We obtained data from a recent patient cohort enrolled in the TRACK-SCI prospective study and observed that despite the high performance of the final $BP_{log}$ and $BP_{XGB}$ models on training data, both performed poorly during validation. Population similarity analysis by PSI revealed that almost every feature exhibited significant data drift, indicating differences between the two patient cohorts. We reviewed the PSI and cluster analysis results with TRACK-SCI clinicians who validated that the observed changes corresponded with evolving guidelines including moving SCI patients into surgery sooner and improving blood pressure management to avoid hypotension [43, 44]. Indeed, both practices were implemented during the prospective era, and the overarching findings illustrate the fact that biomedical data inevitably shifts with ever-updating clinical practices [25]. Furthermore, while all of our data was collected at UCSF, the phenomenon of data drift can apply to data from different medical centers; clinical ML studies commonly utilize data from only a single source, and resultant models fail to generalize. Thus beyond simply validating models with novel data, the analytical framework for successful clinical ML application should also contrast the training and validation populations with quantitative metrics alongside domain expertise to identify critical clinical context to inform model validity.

Understanding population similarity can also provide avenues for improving clinical models: a simplistic approach could be to retrain the model with more recent or representative data while balancing dataset size and population similarity. Researchers could also directly incorporate sources of data drift and population dissimilarity into the modeling strategy, such as via cross-hospital validation, to produce the most generalizable models and inferences [53]. A third possibility is to combine data across time and centers to obtain a larger sample size, thus allowing for a holdout partition that better mimics the training dataset. With careful, balanced representation of patients and feature values in training and holdout partitions, researchers

could identify common predictive features that are generalizable across a broader clinical population. Specific to SCI, harmonizing data across multiple clinical programs could help reinforce future ML studies with a more comprehensive patient dataset. Most importantly, once a model is deployed into clinic, monitoring for data drift will be necessary to determine whether the model will need to be retrained to keep up with changing medical practices.

Especially for datasets with a small observations-to-features ratio as is common in clinical research of rare disorders, parsimony can improve performance by decreasing multicollinearity and removing low-signal features while improving interpretability [54]. We deployed an iterative wrapper approach [35]: backward feature selection based on recalculated pFI rankings to optimize LogLoss. While the process is model-agnostic, the results are affected by the characteristics of the underlying algorithms; the final parsimonious feature lists differed between $BP_{log}$ and $BP_{XGB}$, and $BP_{XGB}$ exhibited in greater pFI instability between each feature reduction step. This can be attributed to how regularized regression and tree-based models handle multicollinearity—a common trait of clinical datasets—which affects pFI as colinear features are eliminated. Critically, there is no definitive approach to feature selection; clinical verification of the final parsimonious feature list is necessary given that the preserved features depend heavily on the implemented reduction method and models [34, 35]. More broadly, pFI has been shown to overestimate the importance of colinear features, especially for tree-based models [55]; future work should refine the framework by employing techniques that better address feature collinearity, such as Shapley additive explanations or accumulated local effects plots [56, 57].

We allowed for preselection of features by domain experts during feature reduction, and because the process inherently optimizes model accuracy, we can determine if preservation of expert-selected features improves or undermines the final model performance. Here, by preserving the time a patient spent below 76 mmHg MAP, we improved the maximum performance for the parsimonious $BP_{XGB}$ model. The process can also be applied to test other hypotheses, including ones where deliberate experimentation might be difficult or impossible. For example, whether a SCI patient receives specific treatments is a matter of clinical care rather than experimental design, but we can compare the impact of excluding or including the treatment feature on the final model accuracy to glean a relationship between treatment and outcome. Such findings would provide further insight to the efficacy of clinical practices that may be difficult to test experimentally and highlight areas for future, targeted clinical research. Importantly, removed variables are not necessarily unimportant or uninformative; feature selection ultimately reflects the representation of samples and the limitations of the dataset. Continual validation and updating of the parsimonious feature list alongside the model are critical for maintaining and improving clinical models. This aligns with the budding concept of expert-augmented AI wherein the interaction between human expertise and ML leads to better models.

Our results also provide more granularity on how intraoperative MAP thresholds relate to outcome: the time patients spent outside the upper MAP threshold contributed more to model prediction than the time spent below the lower threshold. Additionally, the corresponding PDPs reveal that the critical out-of-threshold duration for worse outcome is shorter for hypertension (> ~70 min) than for hypotension (> ~150 min). Previous blood pressure management studies have primarily focused on hypotension as a contributor to worse patient outcome [58–60] despite the increased risk of cardiovascular and cerebrovascular complications as a result of hypertension [61]. Notably, the importance of perioperative hypertension for SCI outcome has been observed both clinically and preclinically [26–28, 62]; this is the first analysis to suggest that hypertension is more predictive of worse outcome than hypotension, thus proposing that careful MAP management should strive to avoid hypertension while

minimizing hypotension. Future prospective clinical studies should extend the verifiability of the findings throughout broader SCI patient care such as during treatment in the emergency room and intensive care unit. Moreover, the illustrated framework can be similarly applied to investigate other modifiable components of clinical care as well as identify predictors of other patient outcome metrics including chronic recovery.

Ultimately, there remains significant untapped potential for AI-driven impact on clinical practices, precision medicine, and general patient care, especially for rarer diseases and disorders. However, the challenges of achieving inferential reproducibility with AI/ML dictate the need for parallel development of explainable and reproducible AI methods that augment powerful processes such as AutoML. By unboxing even the most complex models and directly empowering clinical experts to guide the modeling process, we can build an essential culture of trust for AI in biomedicine towards a bidirectional relationship where clinicians inform AI development and AI applications effectively support clinical decision-making to improve patient care.

## Materials and methods

### Datasets

The data were collected and de-identified by the Transforming Research and Clinical Knowledge for Spinal Cord Injury (TRACK-SCI) program [20] and contains clinical variables (i.e. features) collected during acute hospitalization and SCI-related surgery. The training dataset consisted of 74 clinical records collected between 2005–2011. After implementing our AutoML workflow, we obtained a second dataset for model validation from TRACK-SCI consisting of 59 clinical records collected after 2015. Protocols for data collection and extraction were approved by the Institutional Research Board (IRB) at the University of California, San Francisco under the protocol numbers 11–07639 and 11–06997.

Of note, 18 of the 47 features in the full feature list were derived from time-series data for intraoperative heart rate, systolic blood pressure, diastolic blood pressure, and mean arterial pressure (MAP). Each set of time-series data was summarized as mean, standard deviation, skew, and kurtosis features for each individual patient. Additionally, the total time each patient spent above or below a MAP threshold (starting with 104 and 76 mmHg respectively) was derived from the time-series MAP data. The prediction target *AIS_is_improved* was derived from the patients' AIS scores as assessed by clinicians using the International Standards for Neurological Classification of SCI (ISNCSCI) exam at the time of hospital admission and discharge. Specifically, *AIS_is_improved* was assigned a value of 0 for no improvement or a value of 1 if the patient's AIS score improved between admission (*AIS_ad*) and discharge (*AIS_dis*). Notably, the training data included 39 patients who improved in AIS score while the validation dataset included 14 such patients. The *AIS_dis* feature was excluded for model training since it was used to derive the prediction target and would cause target leakage. The remaining 46 features used for modeling are listed in S1 Table.

### AutoML model generation and feature importance

Among various available implementations of AutoML, we utilized the DataRobot commercial AutoML platform for our workflow [5]. Access and application of the platform was done primarily through the API in Python. Performance values (LogLoss and Area Under Curve; AUC), mutual information between predictors, permutation feature importance (pFI), partial dependence plot (PDP) values, receiver operating characteristic (ROC) values, model validation predictions, estimated best F1 thresholds, and population stability index (PSI) values were downloaded and then analyzed and graphed in R with the *tidyverse* package [63, 64].

The training data was uploaded to the platform which generates a new project instance (i.e. project). Each project encapsulates a specific set of modeling inputs and parameters such as the training data, the type of ML problem (i.e. regression vs classification), partitioning strategy, and others that define the initial state of the AutoML process. For our projects, we assigned distinct random seeds which specifically affected the unique partitioning arrangement of the data for modeling. Given that the AutoML platform requires a minimum of 100 observations to perform automated classification modeling, we accounted for the small number of records in our dataset by duplicating the entry for each patient, thus doubling the dataset from 74 to 148 records. Importantly, we specified a 10-fold cross-validation strategy while ensuring that each of the ten partitions had at least one representation of each of the possible prediction target values and that duplicated records were always partitioned together.

For the first round of modeling, we included 46 features as predictors (S1 Table). The AutoML platform generated 80–90 possible configurations of data preprocessing steps and algorithms (i.e. blueprints, with each blueprint being assigned a unique identification number). Blueprints range from simple (e.g. $BP_{log}$: regularized logistic regression model with a spline transformation preprocessing step) to complex (e.g. $BP_{XGB}$: extreme gradient boosted tree with a modified TensorFlow Variational Autoencoder preprocessing step), and the platform automatically performs data preprocessing and algorithm-specific optimization to maximize the final model performance. To identify the best-performing models, the platform first trained the blueprints on a small subset of the dataset and selected the top performing blueprints according to their validation LogLoss accuracy. The blueprint selection process was repeated with a larger subset of data for a second round of selection. The remaining blueprints, numbering between 30–40 total, were then optimized on the full dataset, and cross-validation accuracy was calculated.

To characterize the stability of the modeling process, we applied the strategy of repeated 10-fold cross-validation with 25 repetitions. Each repetition corresponded to a project with a unique random seed that determined the unique arrangement of data in the partitions. Each project implemented the same blueprints for AutoML; we accordingly aggregated the performances for each blueprint across all 25 projects by calculating the mean and 95% confidence interval for the corresponding cross-validation performances (i.e. performance). We arranged the models according to mean performance and plotted those that outperformed the Majority Class Classifier benchmark model which simply predicts every patient as having improved—the majority class of the *AIS_is_improved* target.

Permutation feature importance (pFI; also termed feature impact on the platform) was calculated through a permutation-based approach by the AutoML platform [33]. In brief, the values of a single feature were permuted, and the resulting loss in LogLoss accuracy was calculated. The permutation and performance loss assessment were repeated multiple times to generate an average accuracy loss. This process was performed on every feature individually. The platform further normalized the pFI values to the maximum pFI value observed; pFI in this study thus refers to the normalized values. We aggregated the pFI values across all 25 projects to calculate the mean pFI and 95% confidence interval for each feature. The features were then arranged from highest to lowest pFI for visualization.

## Performance precision analysis

To characterize the relationship between number of aggregated projects and the precision (i.e. variability) of model performance as a result of different partitioning arrangements, we created 150 projects, each with a unique random seed and corresponding partitioning arrangement. We optimized both $BP_{log}$ and $BP_{XGB}$ in each project with the 46 features using the AutoML

platform. We collected the resulting cross-validation LogLoss and AUC performance values for all 150 projects. We then performed the following sampling analysis:

1. Randomly sample one project.

2. Randomly sample another project without replacement. The newly-sampled project and any previously-sampled projects form the current project aggregate.

3. Calculate the standardized performance CI width with the current project aggregate.

4. Repeat steps 2–3 until all 150 projects have been aggregated.

5. Perform steps 1–4 1000 times.

6. Calculate the expected (i.e. mean) standardized performance CI width with corresponding 95% confidence intervals for each number-of-projects-aggregated.

   Standardized performance CI width was calculated with the following formula:

$$Standardized\ performance\ CI\ width = \frac{Observed\ Confidence\ Interval\ Width}{Mean\ Performance} * 100$$

The results were visualized with an emphasis on the 25-project point. The process was repeated for $BP_{log}$ and $BP_{XGB}$ as well as for LogLoss and AUC metrics.

## Feature instability analysis

To characterize the relationship between number of aggregated projects and *feature instability*, we used the same 150 projects as in the performance precision analysis. For each instance of $BP_{log}$ and $BP_{XGB}$, we calculated the normalized pFI values. To obtain the pFI ranks, we ordered and ranked the pFI values for each specific instance of the model and project from highest to lowest. As a metric for feature instability, we calculated the feature rank instability (FRI):

$$Feature\ Rank\ Instability\ (FRI) = \sum\nolimits_{i=1}^{f} |FI\ rank_{p,i} - FI\ rank_{q,i}|,$$

where $p$ and $q$ represent two different pFI lists, $i$ is the ith feature, and $f$ is the total number of features.

We then performed a sampling analysis as follows:

1. Randomly sample one project.

2. Randomly sample another project without replacement. The newly sampled project and any previously sampled projects form the current project aggregate.

3. Rank the features according to the mean pFI values in the current project aggregate.

4. Calculate FRI between the pFI lists from the previous project aggregate and the current project aggregate. If only two projects have been aggregated, FRI is calculated between the pFI list from the first sampled project and the pFI list from the current project aggregate.

5. Repeat steps 2–3 until all 150 projects have been aggregated.

6. Perform steps 1–5 1000 times.

7. Calculate the expected (i.e. mean) FRI with corresponding 95% confidence intervals for each number-of-projects-aggregated.

The results were visualized with an emphasis on the 25-project point.

We also performed the sampling analysis on just the ranking of the bottom five features by pFI (i.e. least important features). In this case, the entire feature list was ranked as before, but the FRI was only calculated for the five least important features based on the aggregate with fewer projects (i.e. when comparing 3-project aggregate vs 4-project aggregate, we considered the bottom 5 features from the pFI values of the 3-project aggregate).

To investigate the feature instability during feature reduction for $BP_{log}$ and $BP_{XGB}$, we applied the FRI quantification to compare the feature list before and after each reduction step. Specifically, we calculated FRI for the features that remained after elimination. For example, at feature list size = 41, we calculate the FRI for the 41 features by comparing their rankings between the 46-feature model (before reduction) and the 41-feature model (after reduction).

## Automated feature reduction

We applied an iterative wrapper feature reduction process implementing backward elimination similar to as historically applied to regression models [34]. Notably, the lowest-ranking features by pFI were removed; this feature reduction process can be applied to any blueprint on the AutoML platform. The process is as follows:

1. Start with the full feature list.

2. Calculate average pFI values for each feature across the 25 projects.

3. Remove the 5 features with lowest mean pFI values.

4. Optimize new models on remaining features.

5. Repeat steps 2–4 until no features remain.

6. Identify the range of feature list sizes containing the likely maximum performance.

7. Repeat steps 2–4 within the range identified in step 6 and using a step size of 1.

The initial step size of five was chosen to balance for computational time needed to retrain 25 models at each elimination step. By aggregating across the 25 projects, we were able to stabilize the pFI rankings.

Importantly, identifying the final parsimonious feature list was determined directly by the resulting model performances. At each elimination step, the model cross-validation performance was calculated and averaged across projects for comparison. The mean model performance values were used to pinpoint the feature list size range at step 6 as well as identify the final best-performing parsimonious model and feature list.

To test whether preservation of *time_MAP_Avg_below_76* would improve final parsimonious model performance with $BP_{XGB}$, we allowed users to preselect features that the process would never eliminate (equivalent to augmenting feature reduction with expert guidance). If the preselected features landed in the elimination range of the pFI ranking, the process selected the next lowest-importance feature instead. We accordingly selected *time_MAP_Avg_below_76* to be preserved.

## Feature interpretation

The AutoML platform implements partial dependence plots (PDPs) for feature interpretation [40]. In brief, the platform averaged the outcome predictions for the training dataset while converting the values of a single feature to a single value. The set value for the feature of interest was then changed, scanning either across the continuous range or all possible categorical values depending on the feature's data type. Plotting the average outcome prediction by the

possible feature values produced the feature's PDP for the model. We additionally pooled the partial dependence values across the 25 projects, calculated the mean and 95% confidence intervals, and created an aggregated PDP for each feature in the parsimonious $BP_{log}$ and $BP_{XGB}$ models.

## MAP threshold validation

To investigate the MAP thresholds that would be most predictive of patient outcome, we first removed the MAP threshold features from the final parsimonious feature lists of $BP_{log}$ and $BP_{XGB}$. We then created new lists by including a single MAP threshold feature using a different lower (in range of 70–85 mmHg) or upper (in range of 95–115 mmHg) threshold. Sweeping through each possible threshold value, this produced 16 feature lists with a lower MAP threshold feature and 21 feature lists with an upper MAP threshold feature. We additionally included a feature list with no MAP threshold feature. Across the 25 projects, we optimized models for each feature list, aggregated the model performance values, and summarized and plotted the results as mean and 95% confidence intervals. The model performance for the feature lists including both a lower and upper MAP threshold feature was the resulting parsimonious model from the feature reduction process prior.

## Model validation

To validate the parsimonious $BP_{log}$ and $BP_{XGB}$ models, we uploaded the validation dataset to the AutoML platform and predicted the probability of AIS improvement for each patient. The AutoML platform also calculated the best F1 threshold—the value that maximizes the F1 score —for each model in each project. We aggregated the predictions for each patient across the 25 projects to calculate mean and 95% confidence intervals. We similarly summarized the best F1 threshold values. To produce the confusion matrices, we compared the mean prediction value for each patient against the mean best F1 threshold value. Mean prediction values above the mean F1 threshold were considered positive predictions (i.e. patient improved) and conversely for negative predictions (i.e. no improvement).

To determine whether there is data drift between the training and validation dataset, we deployed the parsimonious models on the DataRobot servers to access the data drift feature. In brief, the platform determines data drift between training and validation datasets by calculating the population stability index (PSI) for each of the features [42].

Combining both the training and validation datasets, we additionally performed dimensionality reduction via UMAP (*umap* R package [65]) for the 15 features preserved in the parsimonious $BP_{log}$ and $BP_{XGB}$ models. Importantly, 9 of the 133 samples were missing values and were thus removed via listwise deletion for clustering analysis prior to UMAP. The resulting UMAP scores were used to cluster the patients via HDB Clustering (*dbscan* R package [66]) with a minimum cluster size of 8. The datapoints were then grouped according to training vs validation dataset and plotted. The circular borders containing the clusters were drawn manually for visual clarity. For the numeric features, we calculated the mean and 95% confidence interval of the distribution within each cluster.

## Supporting information

**S1 Fig. Normalized permutation feature importance (pFI) of each feature, aggregated from the 25 projects.** (A) Of note, $BP_{log}$ ranked the *time_MAP_Avg_below_76* and *time_MAP_Avg_above_104* highest. (B) Conversely, the two MAP threshold-related features were ranked 11th and 18th in pFI by $BP_{XGB}$. The majority of high pFI features across both models were features derived from the intraoperative timeseries data for heart rate, diastolic blood

pressure, systolic blood pressure, and mean arterial pressure (MAP). Both models also highly ranked a feature encoding initial injury severity: *MRI_1_BASIC_Score* for $BP_{log}$ and *AIS_ad* for $BP_{XGB}$.
(TIF)

**S2 Fig. Standardized performance precision analysis as a function of number of projects aggregated.** As the number of projects increased, the performance precision improved (i.e. standardized performance CI width decreased). (A, B) By LogLoss, $BP_{log}$ started with a standardized performance precision of 5.22 ± 0.24% with 2-project aggregation and decreased to an average of 1.85 ± 0.01% with 25-project aggregation (A). By AUC, $BP_{log}$ started with a performance precision of 8.03 ± 0.38% and decreased to an average of 2.79 ± 0.02% when aggregating 25 projects. (C, D) Similarly by LogLoss, $BP_{XGB}$ started with a standardized performance precision of 5.42 ± 0.32% and decreased to an average of 2.06 ± 0.04% at 25 projects (C). By AUC, $BP_{XGB}$ started with a performance precision of 8.36 ± 0.38% and decreased to an average of 2.84 ± 0.02% when aggregating 25 projects.
(TIF)

**S3 Fig. AUC performances for the feature reduction process.** (A) Feature reduction of $BP_{log}$ showed maximum AUC at the 8-feature parsimonious feature list (AUC = 0.84 ± 0.02). The 9-feature parsimonious feature list had an AUC of 0.83 ± 0.02. (B) Feature reduction of $BP_{XGB}$ showed maximum AUC at the 9-feature parsimonious feature list (AUC = 0.87 ± 0.01). The 11-feature parsimonious feature list had a similar AUC of 0.87 ± 0.01.
(TIF)

**S4 Fig. Feature instability analysis for (A) $BP_{log}$ and (B) $BP_{XGB}$ during backward feature reduction process.** FRI was calculated by comparing the pFI ranking before and after each feature reduction step and only summing the features that appeared in both lists (i.e. features that were not removed at the step). Notably, $BP_{XGB}$ exhibited higher FRI at each step than for $BP_{log}$; elimination of features resulted in more shifting of features by pFI rank for $BP_{XGB}$.
(TIF)

**S5 Fig. Partial dependent plots (PDPs) for additional features from the best-performing parsimonious feature list for $BP_{log}$.** In order of highest pFI to lowest: (A) *DiaBP_skew*, (B) *HR_kurtosis*, (C) *SysBP_sd*, (D) *MAP_kurtosis*, (E) *HR_sd*, and (F) *Time_to_OR_a*. PDPs of *MRI_1_BASIC_Score*, *time_MAP_Avg_above_104*, and *time_MAP_Avg_below_76* are shown in Fig 4.
(TIF)

**S6 Fig. Partial dependent plots (PDPs) for additional features from the best-performing parsimonious feature list for $BP_{XGB}$.** In order of highest pFI to lowest: (A) *MAP_kurtosis*, (B) *DiaBP_skew*, (C) *HR_sd*, (D) *Cervical_Injury*, (E) *TBI_Present*, (F) *HR_mean*, (G) *Vertebral_Artery_Injury*, and (H) *MAP_mean*. PDPs of *AIS_ad*, *time_MAP_Avg_above_104*, and *time_MAP_Avg_below_76* are shown in Fig 4.
(TIF)

**S7 Fig. AUC performance plots for investigating lower and upper MAP thresholds using best-performing parsimonious $BP_{log}$ and $BP_{XGB}$ models.** (A) Similar to the LogLoss plots, the best-performing lower threshold values were 74, 75, 76, and 79 mmHg and the best-performing upper threshold values were 103, 104, and 105 mmHg for $BP_{log}$. Of the best-performing thresholds, inclusion of an upper threshold features produced greater improvement to AUC than inclusion of an individual lower threshold feature. (B) For $BP_{XGB}$, the best-performing lower threshold values were 74, 75, and 76 mmHg, and the best-performing upper

threshold values were 103 and 104 mmHg. Similar to $BP_{log}$, of the best-performing thresholds, inclusion of an individual upper threshold feature improved AUC performance more than inclusion of an individual lower threshold feature.
(TIF)

**S8 Fig. Receiver operating characteristic (ROC) curves of individual projects and the averaged curve showing improvement in prediction performance through the workflow.** (A) ROC curves of the L2 regularized linear regression model $BP_{log}$ trained on the initial feature list with the exclusion of the MAP threshold features. The average model AUC was 0.63 ± 0.02. (B) ROC curves of $BP_{log}$ trained on the full feature list including the two MAP threshold features. The average AUC was 0.68 ± 0.02. (C) ROC curves after performing feature reduction with $BP_{log}$ to find the best-performing parsimonious model (9-feature parsimonious feature list). The average AUC increased to 0.84 ± 0.02. (D) ROC curves after testing different MAP thresholds with $BP_{log}$ and selecting for the best-performing lower (79 mmHg) and upper (104 mmHg) thresholds. The resulting AUC improved incrementally (AUC 0.85 ± 0.02) compared to using 76 mmHg and 104 mmHg. (E) ROC curves after performing the workflow on the eXtreme gradient boosted tree model $BP_{XGB}$. The parsimonious feature list consisted of 11 features and the best-performing MAP thresholds were 76 and 104 mmHg. The average model AUC was 0.87 ± 0.01.
(TIF)

**S9 Fig. Model validation plots from performing predictions across the 25 projects with a validation cohort of 59 patients.** Of these, 14 patients improved in AIS score while 45 patients did not. Best F1 thresholds as calculated by the AutoML platform were also aggregated from each project (shown in red). (A) Prediction for each validation subject by $BP_{log}$. The average best F1 threshold is 0.41 ± 0.04. (B) Prediction for each validation subject by $BP_{XGB}$. The average best F1 threshold is 0.46 ± 0.04.
(TIF)

**S1 Table. Features and definitions for the 46 features used for modeling.**
(TIF)

**S2 Table. Population stability index (PSI) of the parsimonious $BP_{log}$ and $BP_{XGB}$ model features.**
(TIF)

**S3 Table. Within-cluster mean and 95% confidence interval of numeric features.**
(TIF)

## Acknowledgments

**TRACK-SCI Consortia authorship in alphabetic order**: Beattie MS, Bresnahan JC, Burke JF, Chou A, de Almeida CA, Dhall SS, DiGiorgio AM, Duong-Fernandez X, Ferguson AR, Haefeli J, Hemmerle DD, Huie JR, Kyritsis N, Manley GT, Moncivais S, Omondi C, Pan JZ, Pascual LU, Singh V, Talbott JF, Thomas LH, Torres-Espin A, Weinstein P, Whetstone WD.

## Author Contributions

**Conceptualization:** Austin Chou, Abel Torres-Espin, Nikos Kyritsis, J. Russell Huie, Lisa U. Pascual, Edilberto Amorim, Philip R. Weinstein, Geoffrey T. Manley, Sanjay S. Dhall, Jonathan Z. Pan, Jacqueline C. Bresnahan, Michael S. Beattie, William D. Whetstone, Adam R. Ferguson.

**Data curation:** Austin Chou, Abel Torres-Espin, Jonathan Z. Pan.

**Formal analysis:** Austin Chou, Abel Torres-Espin, Nikos Kyritsis, J. Russell Huie, Adam R. Ferguson.

**Funding acquisition:** Austin Chou, Geoffrey T. Manley, Jacqueline C. Bresnahan, Michael S. Beattie, Adam R. Ferguson.

**Investigation:** Austin Chou, Abel Torres-Espin, Lisa U. Pascual, Jonathan Z. Pan, William D. Whetstone.

**Methodology:** Austin Chou, Abel Torres-Espin, Sarah Khatry, Jeremy Funk, Jennifer Hay, Andrew Lofgreen, Rajiv Shah, Chandler McCann, Adam R. Ferguson.

**Project administration:** Adam R. Ferguson.

**Resources:** Sanjay S. Dhall, Jonathan Z. Pan, Michael S. Beattie.

**Software:** Austin Chou, Sarah Khatry, Jeremy Funk, Jennifer Hay, Andrew Lofgreen, Rajiv Shah, Chandler McCann.

**Supervision:** Adam R. Ferguson.

**Validation:** Austin Chou, Abel Torres-Espin.

**Visualization:** Austin Chou.

**Writing – original draft:** Austin Chou.

**Writing – review & editing:** Austin Chou, Abel Torres-Espin, Nikos Kyritsis, J. Russell Huie, Sarah Khatry, Jeremy Funk, Jennifer Hay, Andrew Lofgreen, Rajiv Shah, Chandler McCann, Lisa U. Pascual, Edilberto Amorim, Philip R. Weinstein, Geoffrey T. Manley, Sanjay S. Dhall, Jonathan Z. Pan, Jacqueline C. Bresnahan, Michael S. Beattie, William D. Whetstone, Adam R. Ferguson.

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
