## [Decision Letter · Decision Letter 0]

28 Feb 2022

Expert-augmented automated machine learning optimizes hemodynamic predictors of spinal cord injury outcome

PONE-D-21-39105

Dear Dr. Ferguson,

We’re pleased to inform you that your manuscript has been judged scientifically suitable for publication and will be formally accepted for publication once it meets all outstanding technical requirements.

Kind regards,

Antal Nógrádi, M.D., Ph.D., D.Sc.

Academic Editor

PLOS ONE

1. Thank you for stating the following in the Competing Interests section: 

I have read the journal's policy and the authors of this manuscript have the following competing interests: SK, JS, JH, AL, RS, and CM are current or former employees of DataRobot and own shares of the company. Access to the DataRobot Automated Machine Learning platform was awarded through application and selection by the DataRobot AI for Good program. DataRobot affiliated authors provided editorial contributions during the preparation of the manuscript. 

All other authors have declared that they have no competing interests

Reviewers' comments:

Reviewer's Responses to Questions

**Comments to the Author**

1. Is the manuscript technically sound, and do the data support the conclusions?

Reviewer #1: Yes

2. Has the statistical analysis been performed appropriately and rigorously? 

Reviewer #1: Yes

3. Have the authors made all data underlying the findings in their manuscript fully available?

Reviewer #1: Yes

4. Is the manuscript presented in an intelligible fashion and written in standard English?

Reviewer #1: Yes

5. Review Comments to the Author

Reviewer #1: The authors have demonstrated artificial intelligence and machine learning (AI/ML) can enhance effective clinical strategies in spinal cord injury (SCI). They have shown the possible treatment efficacy of AI/ML in patients with SCI, that can be an important strategy for these patients. The text is written in almost faultless and readable English. The paper is easy to read and a very important topic in the management of patients with SCI.

6. PLOS authors have the option to publish the peer review history of their article (what does this mean?). If published, this will include your full peer review and any attached files.

Reviewer #1: **Yes: **Tomoo Inoue

---

## [Editor Report · Acceptance letter]

30 Mar 2022

PONE-D-21-39105 

Expert-augmented automated machine learning optimizes hemodynamic predictors of spinal cord injury outcome 

Dear Dr. Ferguson:

I'm pleased to inform you that your manuscript has been deemed suitable for publication in PLOS ONE. Congratulations! Your manuscript is now with our production department. 

Kind regards, 

on behalf of

Prof. Antal Nógrádi 

Academic Editor

PLOS ONE